# Optical-resolution functional gastrointestinal photoacoustic endoscopy based on optical heterodyne detection of ultrasound

Yizhi Liang[1], Wubing Fu[1], Qiang Li[1], Xiaolong Chen[1], Huojiao Sun[1], Lidai Wang[2], Long Jin[1] ✉, Wei Huang[3] ✉ & Bai-Ou Guan[1] ✉

Photoacoustic endoscopy shows promise in the detection of gastrointestinal cancer, inflammation, and other lesions. High-resolution endoscopic imaging of the hemodynamic response necessitates a small-sized, high-sensitivity ultrasound sensor. Here, we utilize a laser ultrasound sensor to develop a miniaturized, optical-resolution photoacoustic endoscope. The sensor can boost the acoustic response by a gain factor of $\omega_o/\Omega$ (the frequency ratio of the signal light and measured ultrasound) by measuring the acoustically induced optical phase change. As a result, we achieve a noise-equivalent pressure density (NEPD) below 1.5 mPa·Hz$^{-1/2}$ over the measured range of 5 to 25 MHz. The heterodyne phase detection using dual-frequency laser beams of the sensor can offer resistance to thermal drift and vibrational perturbations. The endoscope is used to in vivo image a rat rectum and visualize the oxygen saturation changes during acute inflammation, which can hardly be observed with other imaging modalities.

Endoscopic technology is irreplaceable in noninvasive detection and diagnosis of gastrointestinal cancer, inflammation, and other lesions[1,2]. Current video endoscopes can only view the surface and superficial morphological changes. Photoacoustic imaging offers a complementary imaging modality for gastroenterology by detecting the ultrasound waves generated by the absorption of laser pulses by biological tissue. It is promising in cancer diagnostics and oncology studies through imaging of tumor-associated hypoxia and angiogenesis[3,4]. Photoacoustic endoscopy (PAE) uses a thin, flexible probe to reach deep interior cavities of humans and animals for minimally invasive imaging of the esophagus, rectum, urinary, and vaginal tracts[5–13]. It can offer high-resolution imaging by focusing and scanning laser and ultrasound beams. A typical PAE system detects laser-induced ultrasound waves using a focused piezoelectric transducer. The acoustic and optical foci are highly overlapped for maximal detection

sensitivity and optimal spatial resolution. Optical ultrasound sensors have recently received increasing interest for PAE use because of their small size and high-sensitivity density (sensitivity per unit area)[14–25]. They can effectively amplify the acoustic response through the modulation of the signal light (Supplementary Note S3), differing from the straightforward acoustic-to-electrical transduction in piezoelectric detection. For example, a tiny acoustic displacement can be translated into a significant change in the optical intensity with an optical resonator by regulating the light reflections in the cavity many times. A microring resonator and a phase-shifted fiber grating were incorporated into PAE for ultrasound detection[14,22]. Nevertheless, the feasibility of optical sensors for in vivo PAE remains a significant challenge[19,26].

In this work, we report the development of a miniature, high-resolution photoacoustic endoscope for gastrointestinal imaging

[1]Guangdong Provincial Key Laboratory of Optical Fiber Sensing and Communications, Institute of Photonics Technology, Jinan University, Guangzhou, China. [2]Department of Biomedical Engineering, City University of Hong Kong, Kowloon, Hong Kong SAR, China. [3]Department of Gastroenterology, the First Affiliated Hospital, Jinan University, Guangzhou, Guangdong, China. ✉e-mail: tjinlong@jnu.edu.cn; thuangw@163.com; tguanbo@jnu.edu.cn

in vivo by using a laser ultrasound sensor. It offers a spatial resolution of 7.4 μm and a probe size of 2 mm. It can visualize the oxygen saturation (sO₂) change during rectal inflammation and provide associated metabolic information. Multifaceted advantages of the laser ultrasound sensor for PAE are showcased: (a) Sensitivity: The sensor can enhance the detection sensitivity by transducing the acoustic displacement into an optical phase change, resulting in 1.5 mPa Hz$^{-1/2}$ noise-equivalent pressure density (NEPD). The sensor enables the identification of the change in the blood absorption spectrum in the hemodynamic response with a lower illumination dosage. (b) Stability:

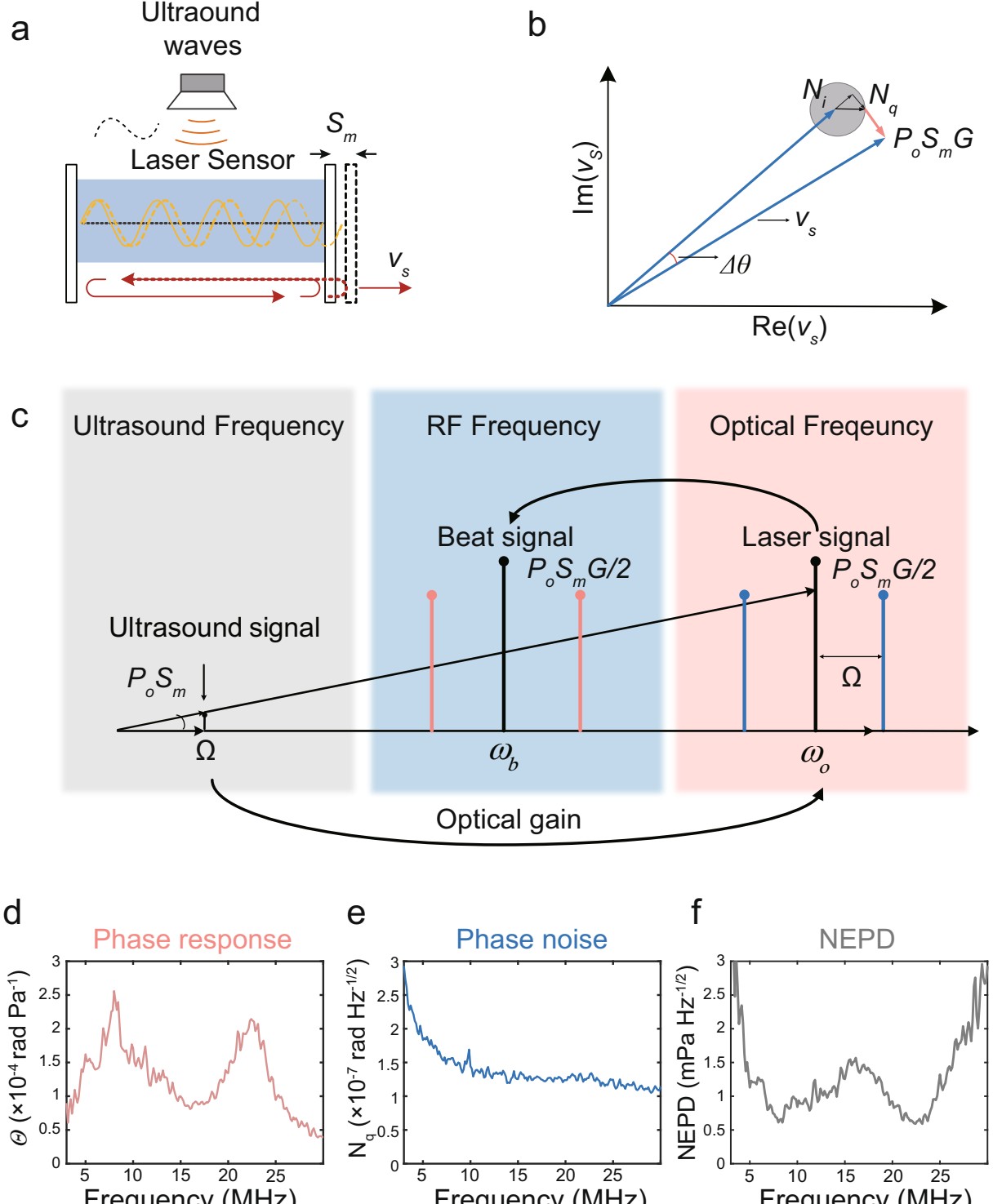

**Fig. 1 | Working principle and characterization of the laser-based ultrasound sensor. a** Schematic of the laser sensor. Ultrasound waves modulate the sensor geometry with a coefficient $S_m$. **b** Vector diagram for measurement of the signal $v_s$ with an acoustically induced phase modulation $\Delta\theta$. The noise (gray shading) is decomposed into the in-phase component $N_i$ and quadrature-phase component $N_q$. **c** Heterodyne phase detection. The acoustic response in laser phase variation is amplified by the gain factor $G$. The induced phase variation is then read out at radio frequencies via heterodyne detection. **d**–**f** Measured phase response, phase noise, and NEPD spectra. RF: radio frequency. NEPD: noise-equivalent pressure density.

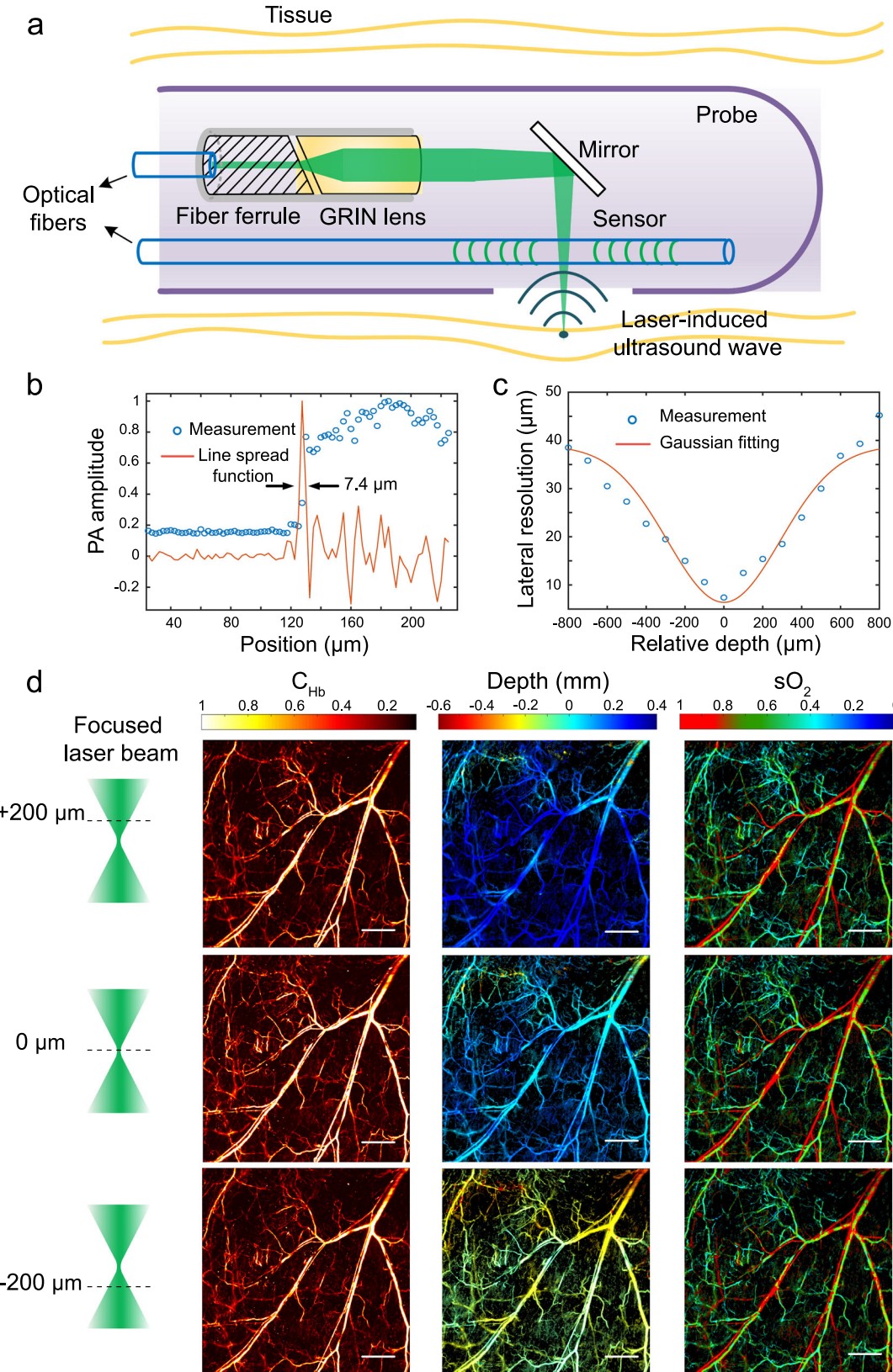

**Fig. 2 | Endoscopic probe. a** Schematic of the probe, which contains an optical guidance fiber capped with a Graded-index (GRIN) lens for excitation light delivery and focusing and a parallel sensor laser for ultrasound detection. **b** Scanning imaging result of a sharp blade at its edge. Selected B-scan image, and extracted line spread function. The estimated lateral resolution is 7.4 μm at the laser spot. **c** Depth-dependent lateral resolution. **d** Scanning imaging results of the hemoglobin concentration ($C_{Hb}$), depth, and oxygen saturation ($sO_2$) of a mouse ear obtained using the probe. The results show almost invariant $sO_2$ quantification results at different working distances. The experiment was repeated six times with similar results. Image sizes: 6 mm by 6 mm. Scale bar: 1 mm.

The common-mode noise cancelation from heterodyne detection stabilizes the sensor output, providing immunity to thermal, vibrational, and rotational perturbations. (c) Miniaturization: The 125-μm diameter sensor may allow a miniature probe size that fits the narrow instrument channel of existing video endoscopes. The laser ultrasound sensor opens an alternative avenue toward the implementation of high performance PAE applications in gastroenterology.

## Results

### Optical detection of ultrasound

The ultrasound sensor for PAE is a short, linear-cavity fiber laser with a round-trip optical path length $L_{opt}$ (Fig. 1a). The ultrasound wave $p$ can modulate the sensor geometry, characterized by the optical path change, expressed as

$$\frac{dL_{opt}}{dp} = S_m \cdot L_{opt} \tag{1}$$

where $S_m$ (Pa$^{-1}$) is a coefficient depending on the geometry and elasticity (see Supplementary Note S2 for details). We write the laser light as a monochromatic wave with unity amplitude as $v_0 = \exp[i(\omega_0 t + \theta_0)]$, where $\theta_0$ denotes the initial phase. The lasing angular frequency $\omega_o$ is defined by the resonance condition $\frac{\omega_o}{c_o} \cdot L_{opt} = m \cdot 2\pi$, where $c_o$ denotes the speed of light in vacuum and $m$ is an integer. Thus, the sensor geometry change induces a lasing frequency variation, expressed by

$$\frac{d\omega_0}{dL_{opt}} = -\frac{\omega_0}{L_{opt}} \tag{2}$$

Based on Eqs. (1), (2) and the phase-frequency relation $\theta(t) = \int \omega_0(t)dt + \theta_0$, the application of a sinusoidal ultrasound wave $p = -p_0\cos(\Omega t)$ to the laser cavity can induce a quadrature-phase modulation $\Delta\theta = p_0 S_m \frac{\omega_0}{\Omega} \sin(\Omega t)$ in the carrier signal[27], as shown in the vector diagram (Fig. 1b), where $p_0$ and $\Omega$ are the acoustic pressure and angular frequency, respectively. Therefore, the modulated signal becomes

$$v_s = v_o \exp(ip_0 S_m G \sin(\Omega t)) \tag{3}$$

Equation (3) suggests that the acoustic response is amplified by a factor $G(\Omega) = \frac{\omega_0}{\Omega}$, leveraging the transduction from the sensor geometry change to the laser phase variation (Fig. 1c). The high contrast between the laser and ultrasound frequencies enables a large gain factor. For example, it has a value of ~2 × 10$^7$ at $\omega_0 = 2\pi \times$ 200 THz and $\Omega = 2\pi \times 10$ MHz. Here, we assume a narrow-band modulation ($\Delta\theta \ll \frac{\pi}{6}$) for simplicity, and only linear modulation is taken into consideration (See Supplementary Note S2 for the case of strong modulation).

We read out the laser phase variation $\Delta\theta(t)$ via heterodyne detection (Fig. 1c). The lasing angular frequencies are $\omega_x$ and $\omega_y$ for the orthogonal polarizations at $\omega_0 = 2\pi \times 193$ THz and are slightly different due to the intrinsic fiber birefringence. They produce a radio-frequency beat signal at the photodetector with $\omega_b = |\omega_x - \omega_y| \approx 2\pi \times 1.74$ GHz, shifting the carrier signal from optical to radio frequencies. The incident ultrasound can vibrate the fiber in a torsional-radial mode, which compresses and stretches the fiber core for the two orthogonal polarizations (see Supplementary Note S2), inducing opposite phase changes $\pm\Delta\theta(t)$. We use the I/Q demodulation method to measure the doubled phase modulation of the heterodyne signal $2\Delta\theta(t)$. By further downshifting the carrier frequency from $\omega_b$ to the baseband and dividing it into two paths with a π/2 phase difference (denoted $I$ and $Q$ in the software-defined microwave devices), we can retrieve the instantaneous phase $arctg(Q/I)$. Figure 1d shows the measured phase response to applied planar-wave ultrasound pulses calibrated with a standard needle hydrophone (see Methods). We

calculated the phase response $\Theta(\Omega) = 2S_m(\Omega) \cdot G(\Omega)$, which agrees with the measured result (Supplementary Note S2). Based on the calculation, the 22-MHz peak corresponds to the mechanical resonance of the silica fiber, enabling a maximum in the $S_m(\Omega)$ spectrum. Multiplying the resonance curve $S_m(\Omega)$ by the gain factor $G(\Omega)$ produces the additional peak at 8 MHz in Fig. 1d because the sensitivity gain effect is more significant at lower frequencies.

However, noise can induce phase measurement uncertainties and limit the detection capability. This noise arises from the sensor laser, photodetector, optical/electrical amplifiers, and data acquisition (DAQ) module. The noise spectral density can be decomposed into in-phase (amplitude noise) and quadrature-phase (phase noise) components $N(\Omega) = N_i(\Omega) + iN_q(\Omega)$ (Fig. 1b)[27,28]. Figure 1e shows the measured phase spectrum $N_q(\Omega)$ of the heterodyne signal, with an average value of $1.5 \times 10^{-7}$ rad Hz$^{-1/2}$. The results in Supplementary Note S1 show that the laser noise dominates below 5 MHz. In contrast, the DAQ noise becomes the main noise source over 8–30 MHz. The thermal noise and shot noise from the photodetector are very weak and can be neglected[28,29]. Notably, the sensor laser has a noticeable intensity fluctuation at approximately 1.7 MHz due to relaxation oscillation. However, the intensity fluctuation can be excluded by the I/Q phase demodulation (Supplementary Note S1) and does not affect ultrasound detection.

Figure 1f shows the NEPD spectrum, which is expressed by

$$NEPD(\Omega) = \frac{N_q(\Omega)}{\Theta(\Omega)} \tag{4}$$

The measured NEPD is below 1.5 mPa · Hz$^{-1/2}$ in the range of 5 to 25 MHz, and the present sensor is comparable to the most sensitive optical ultrasound sensor in the literature. The r.m.s. NEP is 8 Pa in the measured range from 3 to 30 MHz. It is almost two orders of magnitude lower than that of the needle hydrophone (r.m.s. NEP = 768 Pa, NH0200, Precision Acoustics).

The optical heterodyne detection effectively cancels the effect of thermal drift or other low-frequency perturbations by using highly correlated x- and y-polarized laser beams of the sensor. We measured the ultrasound output of the laser sensor with applied thermal drift, fiber bending, and rotational scanning (see Supplementary Note S2). The test results show that common-mode noise cancelation can provide excellent stability, which is beneficial for endoscopic applications with environmental perturbations and fast rotational scanning in PAE. In addition, the heterodyne phase demodulation at radio frequencies does not necessitate any feedback locking. Thus, we have extended the working bandwidth of the frequency demodulation to 200 MHz. With the broadened bandwidth, we achieve a linear response range of 50 kPa (see Supplementary Note S2).

### Endoscopic probe

We then implemented a small-sized photoacoustic probe with the laser sensor (Supplementary Note S5). We encapsulated the sensor and an accompanying light-guiding fiber within a 2 mm diameter medical-grade stainless steel tube (Fig. 2a). The catheter has a side opening to transmit the excitation laser beam and the induced ultrasound waves. The excitation laser pulses are delivered by a single-mode optical fiber and focused by a gradient-index (GRIN) lens before being reflected by a 45-degree prism reflector to the biological tissue. We adjusted the working distance by changing the gap separation between the fiber endface and the lens to fit the cavity to be imaged. Nanosecond 532- and 558-nm lasers were used for photoacoustic excitation (see Methods). The probe can be used to quantify the sO$_2$ level based on the different absorption spectra of deoxy- and oxygenated hemoglobin by recording the photoacoustic signals at these two wavelengths (Supplementary Note S4). This catheter contains no electrical devices or acoustic focusing components, allowing a miniaturized endoscopic

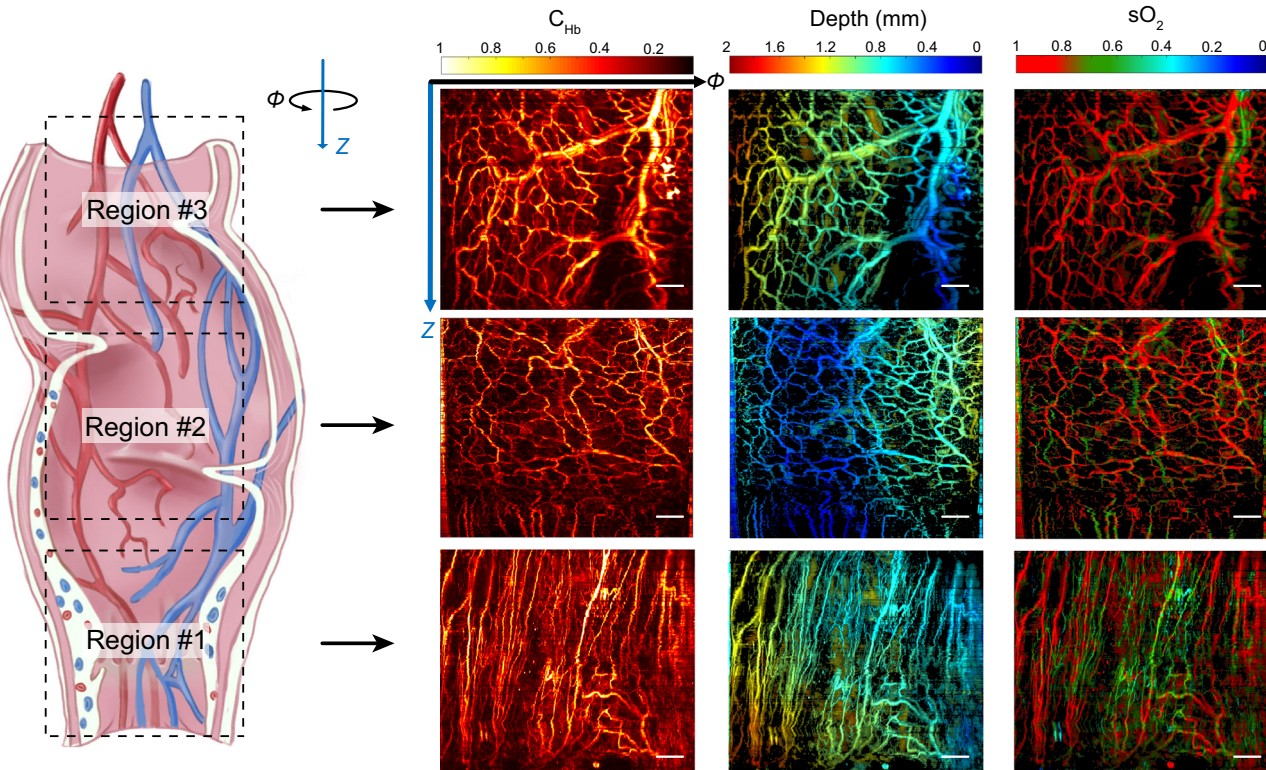

**Fig. 3 | In vivo endoscopy imaging results of the hemoglobin concentration (C_Hb), depth, and oxygen saturation (sO₂) of a rat rectum in three different regions.** The experiment was repeated six times with similar results. Scale bar: 1 mm.

probe. The 0.25 mm diameter of the jacketed optical fiber allows a significant reduction in the probe size. We tested the lateral resolution of the probe by scanning imaging a sharp blade edge (Fig. 2b). The lateral resolution at the focal spot was found to be 7.4 μm for both excitation wavelengths by fitting the line spread functions. In addition, we selected a GRIN lens with a low numerical aperture for an extended depth of view, which allows imaging of multilayer vessels. The test results (Fig. 2c) exhibit a 20 μm resolution at ±300 μm depths from the focal spot and a 30 μm resolution at ±600 μm as a result of wave diffraction.

Notably, the ultrasound sensor works in an unfocused manner, avoiding waveform distortion from acoustic focusing and ensuring a constant sO₂ quantification capability for vessels at different depths (see Supplementary Note S3). We imaged a mouse ear by raster scanning the endoscopic probe at different working distances (Fig. 2d; see "Methods" for details). Here, we obtained hemoglobin concentration (C_Hb) images by plotting the peak-to-peak value of the measured photoacoustic amplitudes. The depth maps were obtained based on the arrival time of the photoacoustic signals. The sO₂ maps can visualize the oxygen consumption from the arteries to the veins and down to the capillary level, enabled by the high sensitivity of the ultrasound sensor. The sO₂ imaging results at depths of −200, 0, and +200 μm demonstrate that the sO₂ quantification is almost independent of the working distance.

### In vivo gastrointestinal endoscopy

To demonstrate the in vivo imaging ability of the photoacoustic probe, we inserted it into a rat rectum and performed rotational scanning endoscopic imaging. Each B-scan contains 2000 A-lines, with an angle step of 0.1 degrees. The B-scan imaging speed was 1 Hz. The corresponding lateral step was 5 μm at the focal plane, with a working distance of 2.75 mm. To achieve volumetric imaging, we recorded B-scan images during constant pullback translation of the probe (10 μm/s) using a motorized translation stage. The laser pulse energy was

250 nJ at both excitation wavelengths, corresponding to an optical fluence below 20 mJ/cm² (ANSI safety limit at the tissue surface). The 1550-nm (193 THz) signal light was transmitted through an optical slip ring, with minimal but unavoidable optical losses and fluctuations in the transmitted intensity. Nevertheless, the heterodyne phase detection was hardly affected by the optical intensity change, and the sensor could have a stable output during the rotational scanning (Supplementary Note S2).

Figure 3 shows photoacoustic images from three different regions of the rat rectum. Region #1 is near the anus, and regions #2 and #3 are located 2 cm and 3 cm deep from the anus, respectively. Each image took a scanning time of ~60 min. The images were processed from a volumetric data set acquired from a ~5.5-mm diameter, 9-mm long, and 210-degree image volume. The PAE results show the high-resolution, three-dimensional vasculature distribution together with the oxygen saturation in the rectum (Fig. 3). The imaging results exhibit significantly different vascular network profiles. Region #1 shows many parallel vessels along the longitudinal direction. Region #2 exhibits relatively uniform, grided vasculature networks. Region #3 shows branch structures, which consist of several trunk vessels surrounded by many branch vessels and capillaries. The sO₂ differences between arteries and veins can be clearly visualized, enabled by the high sensitivity of the laser-based sensor. Compared with the smooth vessels in the ear image in Fig. 2, the rectal vessels in the photoacoustic images present spatial fluctuations and discontinuities due to intestinal peristalsis. Despite this irregular movement, the photoacoustic endoscope can offer a stable output, which guarantees continuous imaging to visualize the hemodynamic response.

### Functional imaging of an inflamed rectum

To demonstrate the ability of the endoscope to offer clinically relevant information, we repeated the rectal imaging during an inflammation process (see Methods). First, the baseline was acquired under a healthy state. Then, after chemically induced acute inflammation, time-lapse

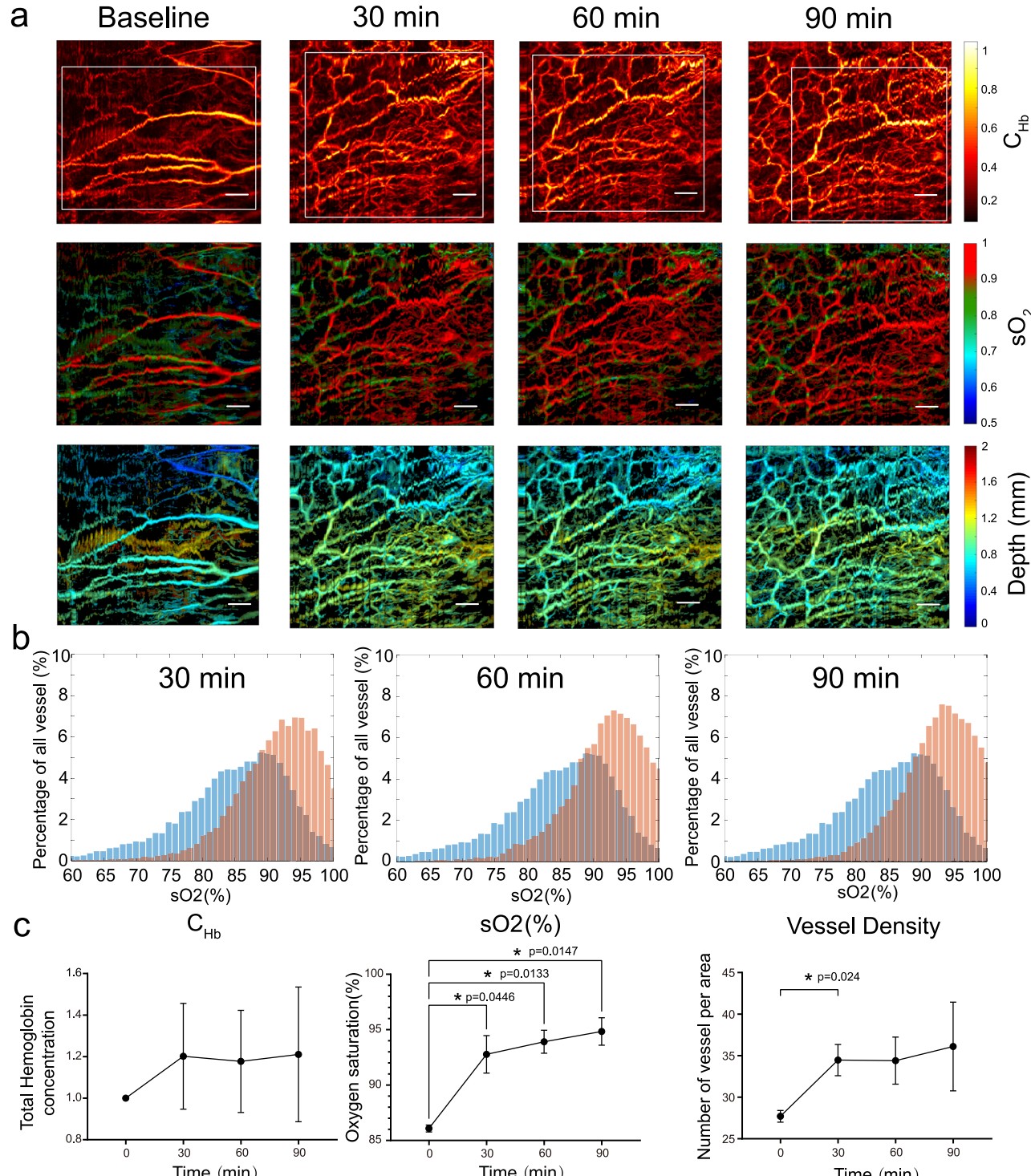

**Fig. 4 | Endoscopic imaging of a rat rectum for visualization of the hemodynamic response during acute inflammation.** **a** Time-lapse images acquired by the photoacoustic endoscope over 90 min. The hemoglobin concentration ($C_{Hb}$), depth, and oxygen saturation ($sO_2$) images are presented. The white boxes represent the same imaging region of interest for the statistics. **b** Oxygen saturation distribution changes with respect to the baseline at 30 min intervals. Blue: baseline; red: result from each time-lapse image. **c** Time-dependent changes in the hemoglobin concentration, oxygen saturation and vessel density. Data are presented as the mean±standard deviation, $n = 3$ biologically independent rats were used, one-way repeated-measures analysis of variance (ANOVA and Dunnett's T3 multiple comparisons test) was used in here. *$p < 0.05$. Scale bar: 0.5 mm.

imaging was performed every 30 min. High-resolution, structural and functional photoacoustic imaging can reveal the hemodynamic response to chemical stimulation. The hemoglobin concentration images (Fig. 4a) demonstrate that the capillaries become more visible due to the hyperemia induced by the inflammation. Based on the depth information (Fig. 4a), hyperemia mainly occurs at a superficial depth of the rectal wall. The $sO_2$ image (Fig. 4a) shows that the average oxygen saturation level significantly rises due to inflammation. Figure 4b extracts the percentages of each $sO_2$ level from Fig. 4a, exhibiting an overall oxygen saturation level rise. The statistical results in

Fig. 4c show that acute inflammation increases the average hemoglobin concentration by at least 20%, and the vessel density increases by more than 30% from the baseline (24% in the first 30 min, $p = 0.024$) (see "Methods" for details). A rapid $sO_2$ increase of approximately 7% in the first 30 min ($p = 0.0446$) and a total 9% increase in 90 min ($p = 0.0147$) after chemical treatment are also presented. The measured results agree with previous reports[30], but are acquired at a much higher spatial resolution. The original trunk arteries become less invisible after inflammation because the surrounding vessel diameters and $sO_2$ values significantly increase. This PAE method can non-invasively visualize richer vascularity and enhanced blood and oxygen supply in a narrow cavity in inflamed tissues, which can hardly be observed with other imaging modalities.

## Discussion

Here, we emphasize the sensitivity-boosting capability of both passive and active (laser) optical sensors (see Supplementary Note S3 for details). The optical sensor imposes acoustic modulation on the sensing light and induces a more significant energy variation at the receiver end. As a result, the acoustic-to-electrical energy transduction ratio can be significantly greater than 1. In contrast, piezoelectric detection has an energy conversion ratio of only $\eta_{pe} = 0.001$ to $0.01$. This explains why a small-sized optical sensor with dimensions down to hundreds of nanometers[18] can possess a sensitivity comparable to those of bulky piezoelectric sensors. The key parameters associated with the ultrasound sensitivities and energy transduction of optical and piezoelectric sensors are summarized in Supplementary Note S3.

The detection capability of the laser ultrasound sensor could be further improved. The acoustic modulation $S_m$ is on the order of only $10^{-6}$, which is typical for silica- or silicon-based sensors. An additional fiber coating could enhance this modulation for acoustic impedance matching. Moreover, the erbium-doped fiber amplifier (EDFA) in our sensing system introduces spontaneous emission noise (at least 3 dB). The EDFA could be replaced with a Raman amplifier to obtain a lower noise figure. An alternative method is to change the host material germanate or silicate glass to achieve an output intensity of up to tens of milliwatts to avoid the need for any additive optical amplification. The DAQ noise can also be reduced by using higher-resolution demodulation components.

The starting point of this work is to exploit an optical sensor for PAE use based on its small size and high sensitivity. Optical sensing offers versatile possibilities for advancing the performance and applications of PAE. For example, the resonator-based optical sensor reads out the acoustically induced optical intensity variation by using continuous-wave probe light whose wavelength is located at the slope of the resonance dip. The corresponding detection sensitivity can be expressed as $NEPD = \left(\frac{4\sqrt{3}N_i}{9Q_c} + \frac{N_q}{G}\right) \cdot \frac{1}{S_m}$, where $Q_c$ represents the quality factor of the resonator (Supplementary Note S3), suggesting that both in-phase noise and quadrature-phase noise could limit the ultrasound sensitivity. Aiming at higher detection sensitivity, recent works on these sensors focus on geometry design for the enhancement of $S_m$. For example, by using polymers with significantly lower rigidities than silica or silicon, microring[15] and Fabry–Perot cavities[16] can have an $S_m$ up to approximately $10^{-4}$. An optomechanical waveguide that guides a significant fraction of the light in an air gap has been employed for acoustic sensing. The waveguiding structure is acoustically transient due to the use of a thin deformable membrane. Thus, its optical path is highly sensitive to ultrasound waves, presenting a two-order enhancement of $S_m$ (to 0.02) and an unprecedented NEPD of $1.3\ mPa \cdot Hz^{-1/2}$. In addition, a noncontact remote sensing approach has been demonstrated by optically detecting the local pressure at the laser focus in a noninterferometric manner[23]. This method allows fast scanning imaging and does not necessitate an ultrasound coupling medium. At the proximal end, pulsed interferometry to interrogate the optical resonators is proposed to avoid feedback locking operation[14].

This sensing scheme also allows the simultaneous interrogation of multiple sensors with different resonance frequencies.

Optical heterodyne detection has been applied in precise measurement and metrology[31–34]. For example, as a fundamental interferometry scheme, a heterodyne interferometer has been used to detect slight displacements. A dual-wavelength beam from a Zeeman laser[31], for example, is delivered into a Michelson interferometer. Movement of one of the mirrors induces a Doppler shift, which is read out via frequency demodulation of the laser heterodyne signal. Another example is the optical frequency comb (OFC), a specialized laser source with evenly spaced teeth in its optical spectrum[35,36]. An OFC measures optical frequencies in the microwave regime via heterodyne detection. For example, dual OFCs with a slight difference in the comb spacing have been used to sense laser-induced acoustic waves for trace gas detection[36]. However, limited in the working bandwidth by the frequency spacing difference, this sensing scheme cannot detect MHz ultrasound yet. The emerging small-sized OFCs based on $Si_3N_4$-based optical resonators hold the potential to be used as acoustic sensors. The basic idea of our sensor design is to realize heterodyne detection with a miniaturized device and a more accessible method. Here, the sensor laser simultaneously acts as a dual-frequency laser source and an ultrasound sensor. The dual-frequency lasing naturally comes from the intrinsic fiber birefringence. This simplified scheme can ease the complexity of the phase reading module and allow common-mode cancelation to resist environmental disturbances.

In summary, via heterodyne phase detection using a laser ultrasound sensor, we demonstrated the implementation of a miniaturized photoacoustic endoscope for in vivo imaging of gastrointestinal hemodynamics. It was used to visualize the hemodynamic response in terms of $sO_2$ changes during inflammation at optical resolution. This approach presented superior imaging capabilities, but the current imaging instrumentation has some limitations. We aim to improve it in the following aspects.

First, the imaging speed is currently limited by the back-and-forth rotational scanning, resulting from the multifiber connection between the probe and the outside instrumentation. A possible solution is to downsize the sensor interrogation module and synchronously rotate it with the probe. This design may allow fast unidirectional rotational scanning with a full-angle view.

Second, the current modality is only capable of photoacoustic imaging. In contrast, the existing PAE systems are capable of photoacoustic/ultrasound dual-modal imaging based on two-way piezoelectric conversion. Therefore, we must use an additional ultrasound-generating optical fiber for supplementary ultrasound endoscopy[37–39]. The current imaging modality will also be integrated with optical coherence tomography or fluorescent imaging to provide multiple contrasts.

Third, the PAE approach could be incorporated with emerging techniques, such as autofocusing or shape adaptation, to cover an extended depth of view[10,12,13]. It can also be extended to intravascular applications to image the plaque lipid in coronary atherosclerosis as a complement to ultrasound endoscopy[40–49] by replacing the hard-tube structure with a soft, flexible probe design to reach deeper cavities.

## Methods

### Sensor fabrication

We fabricated the sensor laser in an active rare-earth-doped optical fiber (EY-305, Coractive Inc.) via ultraviolet inscription of a pair of Bragg gratings. The gratings have reflectivities of 99.9% and 99.5% at 1550 nm and are spaced by 2 mm. When pumped with a customized 980-nm low-noise semiconductor laser (BeoGold Technology), the cavity has an orthogonally polarized laser output. The $x$- and $y$-polarized laser light beams heterodyne at the photodetector after a 45-degree oriented polarizer, producing the radio-frequency beat note.

This signal is then sent for I/Q phase demodulation (see Supplementary Note S1 for details of the ultrasound sensing system).

## Ultrasound measurement and calibration

For sensor characterization, ultrasound pulses were generated by using a piezoelectric transducer (V216, Olympus). The laser-based sensor was 5 mm from the acoustic source. We adjusted its tilt angle to comply with a planar acoustic wavefront. We rotated its principal axis to be consistent with the ultrasound wave incident direction to maximize the acoustic response. We Fourier transformed the time response of the phase variation $d(t)$ under this specific acoustic source to acquire the spectral density $D(\Omega)$.

We calibrated the acoustic response by simultaneously detecting the ultrasound waves with a needle hydrophone. We amplified the measured signals by 41 dB and averaged the pulse time traces $p(t)$ approximately 8000 times to obtain an adequate signal-to-noise ratio. The reference spectral density $P(\Omega)$ (in $Pa \cdot Hz^{-1/2}$) was calculated with the given response $S_n$ = 55 mV MPa$^{-1}$ of the needle hydrophone (averaged over 3–30 MHz). Figure 1d exhibits the calibrated spectral density of the acoustic response $D(\Omega)/P(\Omega)$.

The noise spectrum was acquired from a 2 ms long time trace of the phase variation $n_q(t)$ without any applied ultrasound excitation. The frequency power noise spectral density $|N_q(\Omega)|^2$ (in $rad^2 \cdot Hz$) was estimated by the Welch method with segments of 1400 samples, a Hanning window, and 512 overlapping samples. Figure 1f exhibits the calibrated NEPD spectrum. The r.m.s. pressure was calculated by integrating the NEP over the bandwidth 3–30 MHz as $P_{rms,nep} = \sqrt{\int_{2\pi \times 3\,MHz}^{2\pi \times 30\,MHz} |NEP(\Omega)|^2 d\Omega}$, and the result was 8 Pa.

## Dual-wavelength pulsed laser source and sO$_2$ quantification

For photoacoustic excitation, we used a 532-nm/558-nm dual-wavelength light source. The 558-nm laser beam was produced by optically pumping a single-mode fiber (HB450-SC, Fibercore) with a seed 532-nm nanosecond laser (VPFL-G-20, Spectra-Physics, Inc.). The stimulated Raman scattering (SRS) effect induced energy conversion to the first-order Stokes wave at 545 nm and then to the second-order wave at 558 nm. The maximal pulse energy at 558 nm could reach 1 μJ. We adjusted the polarization state of the incident pump light by using a rotational polarizer to maximize the conversion efficiency. The 558-nm component was then filtered out and combined with another 532-nm laser (AO-S-532, Changchun New Industries) before being injected into the biological tissue. The dual-color pulse trains were interleaved with a time delay of 1500 ns for mouse ear imaging. The time interval was changed to 2300 ns for rectal endoscopic imaging to minimize the effect of unwanted acoustic reflections from the steel tubes. We tested the stability of the dual-color laser output by continuously measuring the pulse energies with a photodetector, and the r.m.s. of each beam was 0.6% and 4%. The excellent stability can minimize the measurement error in the sO$_2$ quantification (see Supplementary Note S4 for more details).

## Depth-invariant sO$_2$ mapping

We imaged the ear of an adult mouse at different working distances using the photoacoustic endoscope. Before imaging, we anesthetized the mouse (BALB/c, male, 2 months old, 25 g in weight) with 1.5% isoflurane. The mouse ear was horizontally placed on a vertical stage and kept stationary during imaging. We mounted the fiber-based photoacoustic probe on a 2-axis linear stage for raster scanning over a horizontal plane. The incident excitation laser beam was vertical to the specimen, and the laser-based sensor detected the photoacoustic waves. The scanning range was 6 mm by 6 mm, with a scanning speed of 5 mm/s. Each image in Fig. 2d contains 1500 by 1500 pixels. We repeated this scanning process to image the same specimen at dif-

ferent working distances by changing the height of the vertical stage. Figure 2d exhibits the depth invariance of the sO$_2$ measurement, although the laser spot varies in diameter at different working distances.

## Animal preparation for endoscopic imaging

We fasted the rat (Sprague Dawley, male, 2 months old, 300 g in weight) for approximately 12 h before the experiment to increase the likelihood of an empty rectum for imaging. Before endoscopic imaging, we anesthetized the animal with 2% isoflurane. Then, we cleaned the rat rectum with a saline laxative enema and injected water into the rectum for acoustic coupling. We maintained the anesthesia level during the imaging procedure by continuously supplying 1.5% isoflurane. All procedures were conducted in "Guiding Principles in the Care and Use of Animals" (GB/T 35892-2018, China) and were approved by the Laboratory Animal Ethics Committee of Guangzhou Huateng Biomedical Technology (IACUC:HTSW220304).

## Imaging of an inflamed rat rectum

We fasted the rat for approximately 12 h before imaging to increase the likelihood of an empty rectum for imaging. To induce acute inflammation, we treated the rat rectum with 10% acetic acid in region #2. We acquired baseline images by endoscopic imaging before treatment. Then, the photoacoustic endoscope was used to repeatedly image the rectal wall in the inflammation region every 30 min.

## Statistical analysis of the inflamed tissue

A vessel segmentation algorithm was used to analyze the hemodynamic changes. Briefly, a thresholding method was used to select the vessel from the background. Then, skeletonization was used to find the middle line of each vessel, which enabled counting of the number of vessels regardless of the size of the vessel. By isolating and finding these vessels, the vessel diameter, C$_{Hb}$, and sO$_2$ in each segment were calculated by using the corresponding values at the middle line of each vessel. The average values of the vessel number, C$_{Hb}$, and sO$_2$ were calculated for each time-lapse image, and the results are shown in Fig. 4b, c. One-way repeated-measures analysis of variance (ANOVA) was used to compare the vascular structures and hemodynamics measured at different time points over 90 min. In all studies, $p < 0.05$ was considered significant. All data are presented as the standard deviation.

## Reporting summary

Further information on research design is available in the Nature Portfolio Reporting Summary linked to this article.

# Data availability

The raw signal and noise data for calibration of the laser sensor, the raw imaging data of animals that were generated in this study have been deposited in the Zenodo database under accession code. https://doi.org/10.5281/zenodo.7246663.

# Code availability

The authors declare that for data collection the commercially available software from LabVIEW 2015 (National Instruments, USA) and Matlab 2018b (Matlab, Mathworks, USA) were used. Data analysis was conducted in Matlab using its built-in functions. The image algorithm has been patented and is available upon discretion from the corresponding author.

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

## Acknowledgements

L.J. acknowledges the financial support from the National Natural Science Foundation of China (62122031), Guangdong Science and Technology Plan (2020A0505100044 and 2020A0505140005). B.-O.G. acknowledges the financial support from the National Natural Science Foundation of China (62135006 and 61860206002), the Local Innovative and Research Teams Project of Guangdong Pearl River Talents Program (2019BT02X105). Y.L. acknowledges the financial support from the National Natural Science Foundation of China (62275104), L.J and W.H acknowledge the Joint Research Scheme of Jinan University.

## Author contributions

L.J., W.H., and B.-O.G. conceived the project. L.J. and B.-O.G. supervised the research. Y.L., W.F., Q.L., and X.C. prepared the sample and performed the experiments. Y.L. and L.W. contributed to the signal demodulation and processing. Y.L., H.S., L.W., and L.J. contributed to data analysis. Y.L. and H.S. performed theoretical analysis. Y.L., L.J., and L.W. prepared the manuscript. W.H. and B.-O.G. participated in the discussion and manuscript writing.

## Competing interests

L.W. has a financial interest in PATech Limited, which, however, did not support this work. The remaining authors declare no competing interests.
