## [Peer Review File · Nature Communications]

Reviewers' Comments:

Reviewer #1:

Remarks to the Author:

Noteworthy results: In this work, the authors use a small-sized laser as the ultrasound sensor to addresses the tradeoff between sensitivity, stability and miniaturization for photoacoustic (PA) endoscopy. They employed heterodyne detection scheme and expertise from fiber optical communication/signal processing to address the sensitivity challenge in the photoacoustic imaging. This sensor is then miniaturized and incorporated into a 2-mm catheter as an endoscopic probe. Then it is used for in vivo gastrointestinal studying and functional imaging of an inflamed rectum.

Significance to the field: The sensitivity of the sensor can be boosted by a gain factor of ratio of optical/ ultrasound frequency (ω_0/Ω), so the resultant noise equivalent pressure density shows an impressive 1.5 mPa Hz^{-1/2}. It is also immune to thermal perturbations and gastrointestinal peristalsis. The authors demonstrated a resolution of $\sim 10\mu\text{m}$ in in-vivo functional photoacoustic endoscopic imaging and mapping of sO₂. The work represents a great step forward in PA endoscopy in terms of sensitivity, resolution and stability against environmental noise/variations, and could be accepted after the authors clarifies the following questions:

1. For in-phase noise (n_i) and quadrature-phase (n_q) components, what sources will cause these two components respectively?
2. To help readers not familiar with the process, please describe "controlled the fiber-lens distance to adjust the focal length" in more detail, as this is an important step to achieve scanning.
3. The sensitivity gain factor is defined as $G=\omega_0/\Omega$, which implies it is inversely proportional to the acoustic frequency. The sensitivity curve 2(e) shows a relatively large peak at ~ 22 MHz, is it due to the sideband in heterodyne scheme rather than gain, right?
4. In line 136, the frequency noise spectrum is defined as $n_f=\Omega\cdot n_q$. Why would it be defined as this way? In the text, it is obtained without applying ultrasound but how does it relate to ultrasound frequency Ω ?
5. In figure 1 (g), Does the signal amplitude mean the beating frequency?
6. Would the laser line width influence the sensitivity? Would the quality factor of the cavity influence the sensitivity for this device?
7. Figure 2(c) shows the resolution degrades with the imaging depth, is this just the result of wave diffraction?
8. For figure s2(b), it will be helpful to briefly describe how the theory curve (dashed gray curve) is obtained. Same comment for equation (S-7) $v=v_0 (1+S_m Q_c \cos(\Omega t)+(n_q G)/Q_c +n_i)$

Reviewer #2:

Remarks to the Author:

The authors reported on an optical-resolution photoacoustic endoscopic device implemented based on the optical heterodyne detection mechanism of ultrasound. To demonstrate its in vivo endoscopic imaging capability, they performed normal and inflamed rat rectum imaging and acquired decent microvasculature (CHb) images and functional maps (sO₂). Overall, this paper earned a positive impression from this reviewer in terms of the optical ultrasound detection mechanism based on optical heterodyne detection; this is the first time this mechanism has been to such a tiny imaging probe (the photoacoustic endoscope), and functional maps of the vasculatures in rat rectums were successfully acquired at a reliable level, even if much lower laser pulse energy was applied. Considering the physical dimensions in which the proposed concepts were successfully demonstrated, this is a nontrivial achievement.

However, when considering the endoscopic device's completeness from the clinical point of view, there is still a long way to go before it can be applied to actual clinical practice because hardware related to endoscopic image acquisition appears to be underdeveloped, requiring further

improvement. Moreover, the presented methods for demonstrating its in vivo imaging capabilities are somewhat obsolete, probably due to the limited hardware capacity. In addition to these technical issues, there are many sentences that were not written thoughtfully and precisely. Most of all, in the introduction and abstract, the authors' story implied the idea that functional imaging, such as sO₂ mapping, based on conventional piezoelectric sensors is almost impossible. This seems to have been written so to highlight the superiority of their optical ultrasound sensor. Although this might be true in an extreme case, this reviewer cannot agree with that current story and thus recommends that the authors rewrite it more accurately and in a more refined manner so that readers are not misled.

1. Except for the part that explains the principle of the optical heterodyne detection of ultrasound, most of the sentences in the introduction and abstract are not logical or accurate. More specifically, the connection between the research background or related specific rationales that the authors present to appeal to the importance of their achievements and the actual related outcomes is logically flawed or based on ideas to which this reviewer cannot agree.

For example, in the abstract (Lines 22–24) and some other portions, the authors stated that the stabilization of the sensor output affected by the heterodyne detection concept can offer a resistance to peristalsis. However, it is somewhat unreasonable to directly connect the two things to highlight the benefit of their sensor because to prevent the adverse effects caused by peristalsis, the image speed is a more important factor that must be considered—in other words, stabilizing the sensor will not directly solve the problem. Furthermore, in this study, the authors only achieved a B-scan image speed of 1 Hz. In Lines 275 and 276, the authors also presented a sentence that takes up a different stance that the rotational speed of a probe is an important factor, as I comment here. Therefore, if the authors want to discuss the problem of signal instability caused by the change in the curvature of the optical fiber due to peristalsis in the mentioned abstract location, it is necessary to describe this in a more accurate way by mentioning related contents more specifically. In relation to this issue, it would be good if the authors presented any supporting data that could substantiate their claim regarding the increased stability benefiting from the heterodyne detection concept.

The authors' unreliable storytelling is also found in many other parts of the manuscript. For example, in Lines 37–45 of the first paragraph of the introduction, the reasons or rationales that the authors present to speak to why endoscopic imaging of the hemodynamic response at optical resolution is challenging are not that persuasive because their reasoning is true only when assuming a somewhat extreme situation. Specifically, the authors wrote the following sentence: "Second, the reduced ultrasound sensitivity can result in inaccurate sO₂ image and even disable functional imaging." Of course, reduced ultrasound sensitivity could eventually cause an inaccurate mapping of sO₂. However, this issue caused by a piezoelectric sensor's poor sensitivity has not really been a concern to this reviewer, who has also been working on photoacoustic imaging. Even when applying a conventional piezoelectric sensor, I do not think it is difficult to achieve the 20 dB SNR that the authors described in Supplementary Note S4. As the authors also know, the more critical factor that affects accuracy is the uncertainty of optical fluence. In this regard, I wonder what efforts the authors made to compensate for the non-uniformity of the optical fluence at different locations when they imaged the rat rectums. I am afraid that the sentence would mislead readers as if it is almost impossible to acquire a reliable sO₂ map if the conventional piezoelectric-based method is applied.

In addition to the above examples, it was also very hard for this reviewer to agree with the sentences listed below.

- (Lines 12 and 13) "However, imaging the hemodynamic response in disease at high spatial resolution is hindered by ultrasound detection capability:" The ultrasound detection capability is also an important factor. However, if expressed in this manner, it would imply that ultrasonic detection capability is the most important factor in achieving the goal.

- (Lines 35 and 36) "This has been fundamentally hindered by the piezoelectric sensors, whose sensitivity quadratically drops with sensor size reduction" I do not think the use of conventional piezoelectric sensors has been the main reason why related probe miniaturization work is difficult because although the authors did not cite these sources, there have already been many reports of

studies that achieved a probe diameter less than 1–2 mm based on a piezoelectric sensing method. Then, through this paper, do the authors want to make the claim that applying the proposed ultrasound sensing mechanism does not involve any difficulty in achieving any required level of miniaturization? The authors need to provide more compelling reasoning.

- Basically, the first sentences in the abstract and introduction follow the same story structure. However, this reviewer could not identify any reports that photoacoustic endoscopically imaged cancer, inflammation, and other lesions from the presented reference list [4–12]. If there are really related reports, it is necessary to cite more relevant or direct instances.

The above are only a few examples, and such inaccuracies in expression occurred in many areas, making it difficult to enumerate them all (in fact, most of the sentences were like that, except for the technical part on the optical detection of ultrasound and the in vivo demonstrations). This reviewer also agrees that developing such an optical ultrasound detector is important for the aimed photoacoustic endoscopic application. However, the authors need to present their claims or rationales more logically and accurately in all sentences in the manuscript using a more persuasive story structure. The current introduction, especially the first paragraph, is just like an introduction suitable for an ordinary journal.

3. The key innovations in this article appear to be ambiguous, and many concepts overlap with those published in previous reports. Compared to previous reports on similar research subjects (also including the former works of the authors such as Liang et al., *Sci Report*, 7:40849, 2017 and Liang et al., *Nature Communications*, 12, 1–10, 2021), what is the main innovation or improvement offered by this work? Is the application of the previously developed fiber sensor technology to photoacoustic endoscopy the main difference? The authors should clarify the novelty of this work more specifically in the authors' response letter, and they also need to make more efforts to make their achievements appear more clearly in the revised manuscript.

4. What do the authors think is the main advantage of the proposed optical sensor over other types of optical ultrasound sensing methods, such as [18–20], from the perspective of "photoacoustic endoscopic application?" Could the heterodyne detection mechanism also be applied to or combined with other types of optical ultrasound sensing mechanisms? Providing related comments briefly in the discussion section (in or around Lines 310–326) would be useful to other researchers.

5. Although the optical sensor is superior to conventional piezo sensors in terms of sensitivity, when considering the 5.5 mm working distance, the ultrasound sensing section with a 12 mm length appears to be too long. I wonder whether there is no undesirable effect caused by the large difference in the arrival times of a spherical acoustic pulse wave originating from a point source to the central and edge parts of the ultrasound sensing section. As photoacoustic signals are generated by a focused laser beam, the approaching acoustic wave is not a plane wave but a spherical wave, which means that the 12-mm-long sensing section will not be located within the wave front created by a single photoacoustic spike at the same time. This reviewer cannot predict whether the two orthogonal modes would be created as desired if two successive pulses (with different acoustic spectra) enter the sensor and be mixed.

In relation to this question, is it, in principle, possible to further reduce the length without affecting the desired performance? It would be better if the authors provided related comments in the discussion.

6. Several reports have already imaged rat colorectums by applying photoacoustic endoscopy. Was it impossible to apply the proposed probe to other organs, such as the esophagus or blood vessels (i.e., through intravascular imaging)? Imaging a rat rectum seems to be relatively easy compared to other endoscopically imageable sites.

7. In relation to the above question, this reviewer also wondered about the physical dimensions of the endoscopic probe, especially the total length of the rotating probe section, to judge how complete the probe's endoscopic imaging capability is. Is the rotating probe section rigid or flexible? Please provide related information in your response letter.

8. According to the description provided in Supplementary Note 5, it seems that the authors performed the required rotational B-scan back and forth. Was this the main reason why only the 210° angular range was imaged in the rat rectum imaging experiments? If so, was the optical slip ring only for the transmission of the 980 nm pump light and the 1550 nm signal light, and were the 532 nm laser pulses for photoacoustic excitation delivered directly to the scanning tip instead of passing through a slip ring? Regardless of the mechanism adopted, it is necessary to describe

related things more clearly and specifically so that other researchers can better understand the related hardware situations. Additionally, it is necessary to comment on how related technical issues can be solved in the future.

9. In Fig. 2(a), the exact configuration between the optical illumination and the acoustic detection parts is unclear. Were the related parts configured so that the 532 nm excitation laser beam can be sent to a target tissue after passing through the optical fiber for acoustic sensing? If so, does no distortion of the 532 nm excitation laser beam occur? In this regard, it would be helpful if the authors presented a related photo in the supplementary material.

10. It was also not specified what plastic tubing or catheter was utilized to image the rat rectums in vivo and how it intervened between the rat rectum and the rotating probe. Was the 2.0 mm diameter for the probe or the entire catheter? Although such a narrow diameter (i.e., the 2.0 mm probe diameter) would be advantageous when it is used in real clinics via an instrument channel, I think it is too narrow to be applied to rat rectum imaging, unlike the authors' claims. Was the plastic tubing that sheathed the rotating probe temporarily placed inside the rectum only when related imaging was performed? If so, how was the related acoustic matching achieved? It would be helpful if the authors presented a related photo in the supplementary material. Furthermore, in the Method d section, is the 300 g weight a possible weight for the one-month-old mouse? Was the specimen not a rat? It is necessary to present related information accurately.

11. In addition, many important specs related to the key optical elements, such as the GRIN lens (e.g., pitch) and optical fiber for the 532 nm laser (e.g., core diameter), were not presented.

12. Supplementary element call-outs, such as those in the Supplementary Notes, are not presented in sequential order.

13. When comparing Fig. 3 and the related statements in Lines 224–226, it seems that the label in Fig. 3 is reversely assigned.

14. It would be useful if the authors specified the approximate pulse widths of the 532 nm input laser pulse and the 558 nm laser wavelength that was converted by the optical fiber. Furthermore, the time delay values presented in Methods c and Supplementary Note S5 are different. What is correct? Please explain.

15. The authors described that a polarization-maintaining fiber was utilized in order to guide the sensing light with minimizing rotation-induced polarization fluctuation. Would it still work normally, although the rotation speed would be increased to a real-time imaging speed (around 30 Hz)? Please comment on this.

16. In Supplementary Fig. S7, why are dot-like signals included?

Reviewer #1

Noteworthy results:

In this work, the authors use a small-sized laser as the ultrasound sensor to address the tradeoff between sensitivity, stability, and miniaturization for photoacoustic (PA) endoscopy. They employed a heterodyne detection scheme and expertise from fiber optical communication/signal processing to address the sensitivity challenge in photoacoustic imaging. This sensor is then miniaturized and incorporated into a 2-mm catheter as an endoscopic probe. Then it is used for in vivo gastrointestinal studying and functional imaging of an inflamed rectum.

Significance to the field: The sensitivity of the sensor can be boosted by a gain factor of the ratio of optical/ ultrasound frequency (ω_0/Ω), so the resultant noise equivalent pressure density shows an impressive 1.5 mPa Hz^{-1/2}. It is also immune to thermal perturbations and gastrointestinal peristalsis. The authors demonstrated a resolution of ~ 10 μm in *in-vivo* functional photoacoustic endoscopic imaging and mapping of sO₂. The work represents a great step forward in PA endoscopy in terms of sensitivity, resolution, and stability against environmental noise/variability. It could be accepted after the authors clarify the following questions:

- ✓ We greatly appreciate your positive comments and constructive suggestions. We have revised the manuscript and the supplementary materials to address your concerns. After reading these comments, we realized the necessity to refine the sensing theory for broad readability. The "optical detection of ultrasound" section in the main text and Supplementary Notes S1 to S3 were correspondingly revised. We classified the questions into two groups: "on the laser sensor" and "on the photoacoustic endoscopy" in the response letter.

Group A. On the laser sensor:

1. (a) The sensitivity gain factor is defined as $G = \omega_0/\Omega$, which implies it is inversely proportional to the acoustic frequency. The sensitivity curve 2(e) shows a relatively large peak at ~ 22 MHz. Is it due to the sideband in the heterodyne scheme rather than gain, right?

(b) For figure s2(b), it will be helpful to briefly describe how the theory curve (dashed gray curve) is obtained.

- ✓ We revised the ultrasound sensing mechanism in the first paragraph, the "Optical detection of ultrasound" section. In this theory, the ultrasound response and the noise are all characterized by phase rather than frequency.

We clarify that the sensor detects acoustic pressure by measuring the induced laser phase change. The acoustic pressure of 1 Pa results in a laser

phase change $S_m \cdot G$ (in rad). In this way, the readers can better understand the amplification mechanism of acoustic response.

We further quantify the acoustic response in Supplementary Note S2. First, we calculated the $S_m(\Omega)$ spectrum by mathematically solving the decoupled Helmholtz equations. The result shows that the 22-MHz peak corresponds to the mechanical resonance, and the $S_m(\Omega)$ spectrum is determined by the silica elasticity and fiber diameter. We then demonstrate the phase response in Fig. 1d and Supplementary Fig. S2. The sensitivity enhancement factor G has a more significant effect at lower frequencies due to its inverse proportion to frequency (Supplementary Fig. S2). Therefore, multiplying $S_m(\Omega)$ with G creates another peak at 8 MHz.

2. In figure 1 (g), does the signal amplitude mean the beating frequency?

✓ In the revised paper, we characterize it as the phase variation as a function of acoustic pressure. We moved the result to Supplementary Note S2.

3. For in-phase noise (n_i) and quadrature-phase (n_q) components, what sources will cause these two components, respectively?

✓ Both in- and quadrature-phase noise components arise from the sensor laser, photodetector, optical and electrical amplifiers, and data acquisition. See paragraph 3, "Optical detection of ultrasound" section. Notably, in the revised paper, the noise spectra were denoted with capitals $N(\Omega)$, $N_i(\Omega)$, and $N_q(\Omega)$.

✓ The relaxation oscillation noise is one of the primary noise sources of the sensor laser. It mainly contributes to amplitude noise (in-phase noise). However, this noise was excluded in the heterodyne phase measurement. See the measured amplitude and phase noise spectra in Supplementary Note S1.

4. Line 136 defines the frequency noise spectrum as $n_f = \Omega n_q$. Why would it be described this way? In the text, it is obtained without applying ultrasound, but how does it relate to ultrasound frequency Ω ?

✓ The phase of a harmonic signal can write as $\theta(t) = \int \omega_0(t) dt + \theta_0$, where θ_0 denotes the initial phase. Taking a Fourier transform yields the relation between the phase and frequency fluctuations $N_f = \Omega N_q$.

✓ In the revised manuscript, we remove that expression and describe all the noises as phase variations to avoid misleading.

5. Would the laser linewidth influence the sensitivity?

✓ Linewidth is a simplified measure of laser noise, and the optical/electrical spectrum in Figure S1b is insufficient for noise characterization. More precisely, the laser noise should be measured in amplitude noise and phase noise spectra.

- ✓ In Supplementary Note S1, we investigated the noise properties and demonstrated the measured noise power spectra $N_q(\Omega)$ (Fig. 1e), and quantified the detection sensitivity in NEPD spectrum N_q/θ (Fig. 1f). Notably, we also demonstrate the amplitude noise N_i of the laser sensor in supplementary Note S1. The noise and NEPD characterization result suggests that N_i does not affect the detection sensitivity in the heterodyne phase demodulation.

6. (1) Would the quality factor of the cavity influence the sensitivity of this device? (2) Describe how equation (S-7) was obtained.

- ✓ The quality factor is essential for passive optical resonators. A high-quality factor can induce a significant optical intensity change in response to ultrasound waves, as described in Supplementary Note S3. In contrast, the laser sensor does not depend on the quality factor. The pump-free-state quality factor of the cavity may affect the noise level but remains a minor effect. We described their different working mechanisms in Supplementary Note S3.

Also, we revised the theory of passive optical sensors, including Equation (S-7). The theory in Supplementary Note S3 suggests that both the resonator and the laser-based optical sensors can amplify the acoustic response. However, their amplification mechanisms and noise properties are somewhat different.

Group B. On the photoacoustic endoscopy:

7. To help readers unfamiliar with the process, please describe “controlled the fiber-lens distance to adjust the focal length” in more detail, as this is an important step to achieve scanning.

- ✓ We have added detailed information on why and how to control the fiber-to-lens distance to adjust the focal length. See Supplementary Note S5. Figure S7d shows the schematic and the calculated result of focal-length tuning.

8. Figure 2(c) shows the resolution degrades with the imaging depth. Is this just the result of wave diffraction?

- ✓ Yes, the resolution degradation with the imaging depth is mainly a result of wave diffraction of a gaussian laser beam. Therefore, we revised the “Endoscopic probe” section and added a schematic showing the focused laser beam and imaging plane in Fig. 2(d). We also added experimental details of how Fig. 2(c) was obtained in Supplementary Note S5.

Reviewer #2

The authors reported on an optical-resolution photoacoustic endoscopic device implemented based on the optical heterodyne detection mechanism of ultrasound. To demonstrate its *in vivo* endoscopic imaging capability, they performed normal and inflamed rat rectum imaging and acquired decent microvasculature (CHb) images and functional maps (sO₂). Overall, this paper earned a positive impression from this reviewer in terms of the optical ultrasound detection mechanism based on optical heterodyne detection; this is the first time this mechanism has been to such a tiny imaging probe (the photoacoustic endoscope), and functional maps of the vasculatures in rat rectums were successfully acquired at a reliable level, even if much lower laser pulse energy was applied. Considering the physical dimensions in which the proposed concepts were successfully demonstrated, this is a nontrivial achievement.

- ✓ We greatly appreciate your positive comments and constructive suggestions on our results. We revised the paper to address your concerns about the significance and novelty of the work, current technological limitations, and PAE experimental details, respectively.

Group A: Significance, novelty, and limitations

1. (a) When considering the endoscopic device's completeness from the clinical point of view, there is still a long way to go before it can be applied to actual clinical practice because hardware related to endoscopic image acquisition appears to be underdeveloped, requiring further improvement. Moreover, the presented methods for demonstrating its *in vivo* imaging capabilities are somewhat obsolete due to the limited hardware capacity.

(b) Several reports have already imaged rat colorectums by applying photoacoustic endoscopy. Was it impossible to apply the proposed probe to other organs, such as the esophagus or blood vessels (i.e., through intravascular imaging)? Imaging a rat rectum seems to be relatively easy compared to other endoscopically imageable sites.

- ✓ We agree with the reviewer on the above points. As clarified in the Abstract and Introduction sections in the revised manuscript, this work showcased the feasibility of using an optical sensor for *in vivo* functional PAI with optical-resolution PAE. However, the imaging instrumentation is still far from clinical application. The "Conclusion and discussion" section discusses the current technological limitations and further improvements.
- ✓ We decided to perform rectal imaging to show the feasibility and advantage of the proposed method, and the readers can compare the result with the previous reports. Furthermore, as the reviewer suggested, we aim to extend its applications to more imageable sites in the esophagus or blood

vessels. See the "Conclusion and discussion" Section.

2. (a) In addition to these technical issues, many sentences were not written thoughtfully and precisely. Most of all, in the introduction and abstract, the authors' story implied the idea that functional imaging, such as sO₂ mapping, based on conventional piezoelectric sensors, is almost impossible. This seems to have been written so to highlight the superiority of their optical ultrasound sensor. Although this might be true in an extreme case, this reviewer cannot agree with the current story and thus recommends that the authors rewrite it more accurately and refined so that readers are not misled. In addition to the above examples, it was also very hard for this reviewer to agree with the sentences listed below.

- (Lines 12 and 13) "However, imaging the hemodynamic response in disease at high spatial resolution is hindered by ultrasound detection capability:" The ultrasound detection capability is also an important factor. However, if expressed in this manner, it would imply that ultrasonic detection capability is the most important factor in achieving the goal.

- (Lines 35 and 36) "This has been fundamentally hindered by the piezoelectric sensors, whose sensitivity quadratically drops with sensor size reduction" I do not think the use of conventional piezoelectric sensors has been the main reason why related probe miniaturization work is difficult because although the authors did not cite these sources, there have already been many reports of studies that achieved a probe diameter less than 1–2 mm based on a piezoelectric sensing method. Then, through this paper, do the authors want to make the claim that applying the proposed ultrasound sensing mechanism does not involve any difficulty in achieving any required level of miniaturization? The authors need to provide more compelling reasoning.

- Basically, the first sentences in the abstract and introduction follow the same story structure. However, this reviewer could not identify any reports that photoacoustic endoscopically imaged cancer, inflammation, and other lesions from the presented reference list [4–12]. If there are really related reports, it is necessary to cite more relevant or direct instances.

-The above are only a few examples, and such inaccuracies in expression occurred in many areas, making it difficult to enumerate them all (in fact, most of the sentences were like that, except for the technical part on the optical detection of ultrasound and the in vivo demonstrations). This reviewer also agrees that developing such an optical ultrasound detector is important for the aimed photoacoustic endoscopic application. However, the authors need to present their claims or rationales more logically and accurately in all sentences in the manuscript using a more persuasive story structure. The current introduction, especially the first paragraph, is just like an introduction suitable for an ordinary journal.

✓ These questions guided us on how to claim the novelty and significance of the work accurately. As a result, we revised the Abstract, Introduction,

Conclusion, and discussion sections to refine the related description. We intend to make the following points here:

a) High-quality PAE is promising for the minimally invasive detection of gastrointestinal lesions. Here we showcase an alternative approach for implementing small-sized, high-resolution PAE using a laser ultrasound sensor via heterodyne phase detection.

b) The laser ultrasound sensor presents multifaceted advantages for PAE use, including miniaturized sensor size, high detection sensitivity, and excellent stability.

c) The fiber-based PAE was used to visualize the hemodynamic response in sO_2 change and provide functional information in the lesion at the optical resolution, complementing the current video endoscopy.

3. Except for the part that explains the principle of the optical heterodyne detection of ultrasound, most of the sentences in the introduction and abstract are not logical or accurate. More specifically, the connection between the research background or related specific rationales that the authors present to appeal to the importance of their achievements and the actual related outcomes is logically flawed or based on ideas to which this reviewer cannot agree.

For example, in the abstract (Lines 22–24) and some other portions, the authors stated that the stabilization of the sensor output affected by the heterodyne detection concept can offer a resistance to peristalsis. However, it is somewhat unreasonable to directly connect the two things to highlight the benefit of their sensor because to prevent the adverse effects caused by peristalsis, the image speed is a more important factor that must be considered—in other word, stabilizing the sensor will not directly solve the problem. Furthermore, in this study, the authors only achieved a B-scan image speed of 1 Hz. In Lines 275 and 276, the authors also presented a sentence that takes up a different stance that the rotational speed of a probe is an important factor, as I comment here. Therefore, if the authors want to discuss the problem of signal instability caused by the change in the curvature of the optical fiber due to peristalsis in the mentioned abstract location, it is necessary to describe this in a more accurate way by mentioning related contents more specifically. In relation to this issue, it would be good if the authors presented any supporting data that could substantiate their claim regarding the increased stability benefiting from the heterodyne detection concept.

✓ We revised the paper according to the above comments and suggestions:

a) The revised paper demonstrates detailed stability test results by measuring the acoustic response under applied thermal drifts, back-and-forth rotation, and fiber bending. The results suggest the resistance to the environmental perturbations, taking advantage of the optical heterodyne detection. See Supplementary Note S2.

b) In the "Conclusion and discussion" section, we discussed that the current back-and-forth rotational scanning limited the imaging speed. Therefore, we are attempting to speed up the imaging by downsizing the optical interrogation unit and letting it rotate with the probe synchronously to enable unidirectional scanning.

4. The authors' unreliable storytelling is also found in many other parts of the manuscript. For example, in Lines 37–45 of the first paragraph of the introduction, the reasons or rationales that the authors present to speak to why endoscopic imaging of the hemodynamic response at optical resolution is challenging are not that persuasive because their reasoning is true only when assuming a somewhat extreme situation. Specifically, the authors wrote the following sentence: "Second, the reduced ultrasound sensitivity can result in inaccurate sO₂ image and even disable functional imaging." Of course, reduced ultrasound sensitivity could eventually cause an inaccurate mapping of sO₂. However, this issue caused by a piezoelectric sensor's poor sensitivity has not really been a concern to this reviewer, who has also been working on photoacoustic imaging. Even when applying a conventional piezoelectric sensor, I do not think it is difficult to achieve the 20 dB SNR that the authors described in Supplementary Note S4. As the authors also know, the more critical factor that affects accuracy is the uncertainty of optical fluence. In this regard, I wonder what efforts the authors made to compensate for the non-uniformity of the optical fluence at different locations when they imaged the rat rectums. I am afraid that the sentence would mislead readers as if it is almost impossible to acquire a reliable sO₂ map if the conventional piezoelectric-based method is applied.

- ✓ We deleted the inappropriate sentences in lines 37 to 45 and refined the Introduction Section to present the current capability of the state-of-art PAE technology, based on the reviewer's suggestions.
- ✓ The uncertainty of optical fluence does affect the accuracy of sO₂ measurement. Therefore, we measured the stability in pulse energy of the dual-wavelength laser source. The uncertainties in pulse energy are only 0.6% and 4.0%, respectively, at 532 and 558 nm. Supplementary Note S5 also gives the corresponding stability test result of photoacoustic signals excited at the two wavelengths. Because of the relatively stable laser output, we can visualize the oxygen-structuration changes caused by the acute inflammation without compensation.
- ✓ Optical scattering in biological tissue may be another influential factor in sO₂ quantification. However, the scattering at superficial depth is relatively weak and was not considered.

5. The key innovations in this article appear to be ambiguous, and many concepts overlap with those published in previous reports. Compared to

previous reports on similar research subjects (also including the former works of the authors such as Liang et al., *Sci Report*, 7:40849, 2017 and Liang et al., *Nature Communications*, 12, 1–10,2021), what is the main innovation or improvement offered by this work? Is the application of the previously developed fiber sensor technology to photoacoustic endoscopy the main difference? The authors should clarify the novelty of this work more specifically in the authors' response letter, and they also need to make more efforts to make their achievements appear more clearly in the revised manuscript.

- ✓ We have refined the claims on the novelty of this work: We use the laser sensor to demonstrate a PAE for *in vivo* imaging of the sO₂ changes in lesions based on the heterodyne phase detection. The optical ultrasound sensor offers an alternative approach toward high-resolution, miniaturized functional PAE. See the Abstract, Introduction, and Conclusion sections.
- ✓ The “*Sci Report*, 7:40849, 2017” work demonstrated the preliminary result in ultrasound detection. It was used for *ex vivo* photoacoustic microscopy under a stationary state. The “*Nature Communications*, 12, 1–10,2021” work used a laser cavity as a medium for optomechanical interaction for acoustic impedance measurement and mapping. This work developed a high-performance PAE by utilizing the laser ultrasound sensor. We made significant progress in the following aspects: First, we achieved an ultrasound detection sensitivity at 1.5 mPa Hz^{1/2}, taking advantage of the sensitivity boosting factor. Second, the sensor enables multispectral photoacoustic imaging to visualize the oxygen saturation at the optical resolution. Third, and most importantly, we found an alternative approach to high-performance PAE and successfully visualized the hemodynamic response in the lesion.

6. What do the authors think is the main advantage of the proposed optical sensor over other types of optical ultrasound sensing methods, such as [18–20], from the perspective of “photoacoustic endoscopic application?” Could the heterodyne detection mechanism also be applied to or combined with other types of optical ultrasound sensing mechanisms? Providing related comments briefly in the discussion section (in or around Lines 310–326) would be useful to other researchers.

- ✓ Compared with the passive optical sensors, the laser sensors are immune to amplitude noise, yielding a more significant sensitivity gain factor. In addition, the heterodyne phase detection avoids any feedback locking and guarantees a stable output. In contrast, the passive optical sensors should balance between high sensitivity and stability, because the optical resonance is also highly susceptible to thermal drift and environmental perturbations, which limits their practical uses in PAE. See the “Conclusion and Discussion” Section and Supplementary Note S3.
- ✓ Optical heterodyne detection is a fundamental method for metrology and

precise measurement in engineering. The basic idea is to measure an optical signal (laser frequency, optical phase...) at radio frequencies. Some emerging photonic devices, for example, silicon-on-insulator frequency combs, may be potentially used for acoustic sensing. This is because their optical outputs are naturally connected to radio-frequency signals. See the "Conclusion and Discussion" Section.

Group B: on the photoacoustic probe and endoscopy

7. Although the optical sensor is superior to conventional piezo sensors in terms of sensitivity, when considering the 5.5 mm working distance, the ultrasound sensing section with a 12 mm length appears to be too long. I wonder whether there is no undesirable effect caused by the large difference in the arrival times of a spherical acoustic pulse wave originating from a point source to the central and edge parts of the ultrasound sensing section. As photoacoustic signals are generated by a focused laser beam, the approaching acoustic wave is not a plane wave but a spherical wave, which means that the 12-mm-long sensing section will not be located within the wave front created by a single photoacoustic spike at the same time. This reviewer cannot predict whether the two orthogonal modes would be created as desired if two successive pulses (with different acoustic spectra) enter the sensor and be mixed.

In relation to this question, is it, in principle, possible to further reduce the length without affecting the desired performance? It would be better if the authors provided related comments in the discussion.

- ✓ The reviewer was right on the acoustic interaction manner with a straight sensor. We added a paragraph in the supplementary material to describe the fabrication of the endoscopic probe. Indeed, only a small fraction (normally incident components) of the spherical acoustic wavefront can effectively induce an optical response. The effect of the oblique acoustic incidence is almost cancelled and will not create any spike-like response. See Supplementary Note S5: A. Endoscopic probe.

Reducing the laser cavity length would surely be beneficial to minimizing the mismatch induced by the spherical wavefront. However, it is difficult because the optical gain of the rare-earth doped fiber is limited. This was also discussed in Supplementary Note S5, described as "*Notably, only the normal incidence components of the spherical ultrasound wave can effectively induce an acoustic response of the sensor. The effects of the oblique components are cancelled due to the phase variation of the acoustic wave along the fiber. As a result, the sensor receives only a small fraction of acoustic energy. The effective interaction length L_{eff} is determined by the acoustic wavelength and the distance to the acoustic source. Despite the unfocused receiving manner, the sensor can provide sufficient sensitivity in PAE. The mismatch between the sensor laser length and L_{eff} can be*

minimized by using shorter laser cavities. Minimization of the cavity length depends on further increasing the optical gain of the rare-earth-doped fiber."

By the way, the laser cavity confined by the two Bragg reflectors is 2 mm long (see Methods), and the length of the sensitive region at half maximum is 2.8 mm, as characterized in Fig. S7e.

8. In relation to the above question, this reviewer also wondered about the physical dimensions of the endoscopic probe, especially the total length of the rotating probe section, to judge how complete the probe's endoscopic imaging capability is. Is the rotating probe section rigid or flexible? Please provide related information in your response letter.

- ✓ We added detailed information on the probe fabrication in Supplementary Note S5: A. Endoscopic probe.

The related description is: "The optical fiber was first enclosed in glass ferrule #1 (inner diameter: 0.2 mm, outer diameter: 0.45 mm) to match the lens diameter. Then, the GRIN lens (GT-LFRL-050-024-20-NC, pitch: 0.24, length 2.94 mm, outer diameter: 0.5 mm, Grintech GmbH) and fiber were aligned and encapsulated in ferrule #2 (inner diameter: 0.5 mm, outer diameter: 0.66 mm) to form the light focusing unit... We then placed a right-angle reflective prism (dimensions: 1 mm by 1 mm) in front of the GRIN lens endface to redirect the focused light beam to the biological tissue... After the WD was determined, all the components in the light focusing unit were fixed by ultraviolet curable adhesive (Optical adhesive 81, Norland). Next, we used another two glass ferrules (inner diameter: 0.3 mm, outer diameter: 0.5 mm, not shown in the figure) to enclose the pigtailed fibers of the sensor laser for structural support, leaving a bare laser cavity for ultrasound detection. The sensor laser, light focusing unit and prism were then fixed in a rigid stainless steel (SUS) tube as an endoscopic probe. The tube had a length of 105 mm and an outer diameter of 2 mm. A 10 mm-long, 180-degree wide window was created to transmit the optical and acoustic beams."

9. According to the description provided in Supplementary Note 5, it seems that the authors performed the required rotational B-scan back and forth. Was this the main reason why only the 210° angular range was imaged in the rat rectum imaging experiments? If so, was the optical slip ring only for the transmission of the 980 nm pump light and the 1550 nm signal light, and were the 532 nm laser pulses for photoacoustic excitation delivered directly to the scanning tip instead of passing through a slip ring? Regardless of the mechanism adopted, it is necessary to describe related things more clearly and specifically so that other researchers can better understand the related hardware situations. Additionally, it is necessary to comment on how related

technical issues can be solved in the future.

- ✓ We added Fig. S8b in Supplementary Note S5 to illustrate the optical connections in the PAE system. The related description is *"As shown in Fig. S8b, a 980/1550 nm WDM and a polarizer were encapsulated in a metal box, which rotated with the endoscopic probe. To minimize polarization fluctuations, we used a polarization-maintaining fiber to guide the signal light (1550 nm). An optical slip ring connected the signal output to the photodetector."*

In addition, we agree with the reviewer on the limited angle of view. The maximal angle-of-view is 285 degrees, as shown in Fig. S8d. We provide experimental details in Supplementary Note S5: B. Photoacoustic endoscopy, and discussed this limitation in the "Conclusion and Discussion" Section.

10. In Fig. 2(a), the exact configuration between the optical illumination and the acoustic detection parts is unclear. Were the related parts configured so that the 532 nm excitation laser beam can be sent to a target tissue after passing through the optical fiber for acoustic sensing? If so, does no distortion of the 532 nm excitation laser beam occur? In this regard, it would be helpful if the authors presented a related photo in the supplementary material.

- ✓ We provide a detailed description of the probe structure in Supplementary Note S5: A. Photoacoustic probe. We used Fig. S7 to illustrate the fabrication, schematic, and photograph of the imaging probe. We also described the optical focusing and optical/acoustic alignment. We could focus the excitation laser beam with minimal distortion using single mode optical fiber and GRIN lens. The related description is *"A polarization-maintaining single-mode optical fiber (HB450-SC, Fibercore, N. A.: 0.14, mode field diameter: 3.5 μm) was used to deliver the excitation laser pulses to the probe. The fiber has a pure silica core to minimize the photodarkening effect... As shown in Fig. S7d, the pulsed laser beam diverges from the fiber endface, passes through an air gap (approximately 0.5 mm), and then converges after passing through the GRIN lens... We then placed a right-angle reflective prism (dimensions: 1 mm by 1 mm) in front of the GRIN lens endface to redirect the focused light beam to the biological tissue."*

11. It was also not specified what plastic tubing or catheter was utilized to image the rat rectums in vivo and how it intervened between the rat rectum and the rotating probe. Was the 2.0 mm diameter for the probe or the entire catheter? Although such a narrow diameter (i.e., the 2.0 mm probe diameter) would be advantageous when it is used in real clinics via an instrument channel,

I think it is too narrow to be applied to rat rectum imaging, unlike the authors' claims. Was the plastic tubing that sheathed the rotating probe temporarily placed inside the rectum only when related imaging was performed? If so, how was the related acoustic matching achieved? It would be helpful if the authors presented a related photo in the supplementary material. Furthermore, in the Method d section, is the 300 g weight a possible weight for the one-month-old mouse? Was the specimen not a rat? It is necessary to present related information accurately.

- ✓ In Supplementary Note S5, we added detailed information of the endoscopic probe and imaging. The related description is *"To avoid direct contact between the rotational scanning probe and biological tissue, we used a stainless-steel tube (inner diameter: 4 mm, outer diameter: 5.5 mm) as a sheath. A side window was produced to allow transmission of the laser beam and photoacoustic signals during the rotational scanning. We covered this window with an acoustically and optically transparent polyethylene terephthalate (PET) membrane. We then filled the sheath tube with deionized water as an acoustic coupling medium."*
- ✓ For the Depth-invariant sO₂ imaging, a mouse ear is used as a sample. The mouse is two-months-old and weight 25 g. The information was corrected in Methods D section. For the endoscopic imaging, the animal model is two-month-old rat with a weight of 300g.

12. In addition, many important specs related to the key optical elements, such as the GRIN lens (e.g., pitch) and optical fiber for the 532 nm laser (e.g., core diameter), were not presented.

- ✓ We provide information on the endoscopic probe in Supplementary Note S5: A. Photoacoustic probe.

The related information is "A polarization-maintaining single-mode optical fiber (HB450-SC, Fibercore, N. A.: 0.14, mode field diameter: 3.5 μm) was used to deliver the excitation laser pulses to the probe... Then, the lens and fiber were encapsulated in ferrule #2 (inner diameter: 0.5 mm, outer diameter: 0.66 mm) and aligned with the GRIN lens (GT-LFRL-050-024-20-NC, pitch: 0.24, length 2.94 mm, outer diameter: 0.5 mm, Grintech GmbH) to form the light focusing unit."

13. (a) Supplementary element call-outs, such as those in the Supplementary Notes, are not presented in sequential order.

(b) When comparing Fig. 3 and the related statements in Lines 224–226, it seems that the label in Fig. 3 is reversely assigned.

- ✓ We have corrected the call-out sequences in the Supplementary notes and Fig. 3.

14. It would be useful if the authors specified the approximate pulse widths of the 532 nm input laser pulse and the 558 nm laser wavelength that was converted by the optical fiber. Furthermore, the time delay values presented in Methods c and Supplementary Note S5 are different. What is correct? Please explain.

- ✓ Supplementary Note S5 gives the measured waveforms of the dual-wavelength laser pulses in Fig. S8f and S8g. The measured pulse widths are 4.7 and 5.0 ns, respectively. The stability test results of the laser-induced ultrasound signals are also given in Fig. S8h. The related description is "*Figure S8f and g shows the waveforms of the 532-nm and 558-nm laser pulses, measured by using a high-speed photodetector (DET025A, Thorlabs) with a 1.8-GHz sampling rate. The pulse widths are 4.7 and 5.0 ns, respectively. The 558-nm pulse waveform changes to a flat-top or a dual-peak profile in the nonlinear wavelength conversion process 10. However, this variation induces an amplitude change in the high-frequency components of the photoacoustic signal, which is hardly detectable by the laser sensor. As a result, the sO₂ measurement is hardly affected. Figure S8h shows the stability test result of the photoacoustic signals excited by the two wavelengths over 30 min. Although the 558-nm laser output presents a more noticeable amplitude fluctuation, it is still acceptable for sO₂ quantification.*"
- ✓ We have checked the delay times. The dual-color pulse trains were interleaved with a time delay of 1500 ns for the mouse ear image and 2300 ns for the rat rectal image. See Subsection c, Methods.

15. The authors described that a polarization-maintaining fiber was utilized in order to guide the sensing light with minimizing rotation-induced polarization fluctuation. Would it still work normally, although the rotation speed would be increased to a real-time imaging speed (around 30 Hz)? Please comment on this.

- ✓ Supplementary Note S2 presents the stability test result when rotating the sensor. The result suggests that the polarization-maintaining fiber can effectively minimize the effect of polarization fluctuation. Based on the result, we are confident that the sensor can still work with a higher rotation rate.

16. In Supplementary Fig. S7, why are dot-like signals included?

- ✓ We checked the data of in vivo imaging, and dot-like signals may be some

unknown absorbers at the surface of the rectum wall. Such unwanted signals also appeared in Fig. 3 in the main text. In comparison, the phantom and stability results did not show similar dots, indicating that it was not a result of the sensor noise. See “*In vivo* gastrointestinal endoscopy” sections.

Reviewers' Comments:

Reviewer #1:

Remarks to the Author:

I am satisfied with the authors response and the revised manuscript significantly improved, and I recommend acceptance.

Just a minor suggestion: much of the terminology used are from optical communication, signal processing fields, and could be difficult for readers in the biomedical ultrasound imaging or photoacoustic imaging fields. So it will be most helpful to use plain terms or describe a bit more the specialized terms to make it easier to read and appreciate the wonderful results obtained using the new scheme.

Reviewer #2:

Remarks to the Author:

Overall, the story in the abstract, introduction, and discussion have been improved compared to the previous version. However, still there were many sentences or expressions that need to be amended (at least those should be reconsidered by the authors). Followings are the ones that I found.

1. In Line 21, it is necessary to reconsider the words "mechanical scanning". If you go with the present expression, it becomes that the ultimate goal of the presented technology to prevent users from "mechanical scanning". Thus, this reviewer recommends the authors to find other more proper expression, such as "mechanical disturbance".

2. This reviewer thinks that the authors need to refine the sentence in Line 27, "However, many early-stage tumors do not produce surface morphological changes but can induce hemodynamic abnormalities". Should it have been written in a structure of "not only~, but also~"? Is the present expression just the thing that the authors really intended? If so, this reviewer would like to suggest the authors to think again about whether an early-stage tumor really does not cause any morphological changes.

3. In Line 36, this reviewer recommends to delete "dual-element" because it is not an essential requirement.

4. How about creating a new paragraph on line 12?

5. In the sentence in Line 40, for the word "light", this reviewer recommends the authors to find some more fit and specific expression instead of going with the simple and general wording "light" because photoacoustic process also uses "light". How about "readout light", for example? Or "signal light"? as you used. If you decide to go with any altered expression anyway, it is necessary to change corresponding ones consistently in other places.

In addition, this reviewer also wonders whether using the word "amplify" could be the best way to precisely depict the process. The authors also need to check many other places where the same word "amplify" are used. That might be the correct expression eventually. However, this reviewer recommends the author to think again.

6. The authors apply inconsistent wording for a same thing. For example, they used "laser sensor" and "optical sensor" mixedly.

7. This reviewer recommends the authors to move the new insertion in Line 206 to a proper place in a supplementary figure because at the current location, it breaks the flow of more important story.

8. I wonder whether the section head "Conclusion and discussion" is reasonable. As far as this reviewer knows, "discussion-conclusion" or "summary-discussion" are more general structures

when wrapping up a paper. Do you think suggesting a discussion after you already making a conclusion is really reasonable? In the current story elements, I think just going with "Discussion" looks enough.

In terms of this kind of structurizing issue, this reviewer also suggests the authors to consider to move the story beginning from Line 246 to Line 261 to the end of the discussion section because it is related to a future study. Of course, this suggestion is not mandatory at all.

This reviewer cannot enumerate all the details of the poorness that the current manuscript includes. Thus, the authors must carefully read the manuscript multiple times and remove such illogical and unprofessional expressions.

Reviewer #1

I am satisfied with the authors' response, and the revised manuscript significantly improved, and I recommend acceptance. Just a minor suggestion: much of the terminology used is from optical communication and signal processing fields and could be difficult for readers in the biomedical ultrasound imaging or photoacoustic fields. So it will be most helpful to use plain terms or describe more specialized terms to make it easier to read and appreciate the wonderful results obtained using the new scheme.

- ✓ We greatly appreciate the positive comments from the reviewer. We have been trying to balance profession and general readability in manuscript writing. We further optimized the presentation in Figs. 1 a to c to help understand the sensing mechanism. The supplementary material contains more professional descriptions for the readers with more knowledge of communications and signal processing.

Reviewer #2

Overall, the story in the abstract, introduction, and discussion has been improved compared to the previous version. However, many sentences or expressions still need to be amended (at least those should be reconsidered by the authors). The followings are the ones that I found.

1. In Line 21, it is necessary to reconsider the words "mechanical scanning". If you go with the present expression, it becomes that the ultimate goal of the presented technology to prevent users from "mechanical scanning". Thus, this reviewer recommends the authors to find other more proper expression, such as "mechanical disturbance".

- ✓ We changed "mechanical scanning" to "vibrational perturbations", as suggested.

2. This reviewer thinks that the authors need to refine the sentence in Line 27, "However, many early-stage tumors do not produce surface morphological changes but can induce hemodynamic abnormalities". Should it have been written in a structure of "not only~, but also~"? Is the present expression just the thing that the authors really intended? If so, this reviewer would like to suggest the authors to think again about whether an early-stage tumor really does not cause any morphological changes.

- ✓ According to the reviewer's suggestions, we revised this part as follows: "Current video endoscopes can only view the surface and superficial morphological changes. Photoacoustic imaging offers a complementary imaging modality for gastroenterology by detecting the ultrasound waves generated by absorption of laser pulses by biological tissue."

3. In Line 36, this reviewer recommends to delete "dual-element" because it is

not an essential requirement.

✓ The words “dual-element” was deleted.

4. How about creating a new paragraph on line 12?

✓ We are afraid that the abstract should be one paragraph, according to the editorial policy of Nature Communications.

5. In the sentence in Line 40, for the word “light”, this reviewer recommends the authors to find some more fit and specific expression instead of going with the simple and general wording “light” because photoacoustic process also uses “light”. How about “readout light”, for example? Or “signal light”? as you used. If you decide to go with any altered expression anyway, it is necessary to change corresponding ones consistently in other places.

In addition, this reviewer also wonders whether using the word “amplify” could be the best way to precisely depict the process. The authors also need to check many other places where the same word “amplify” are used. That might be the correct expression eventually. However, this reviewer recommends the author to think again.

✓ According to the comment, we changed “... use light to amplify” to a more specific expression, as “They can effectively amplify the acoustic response by inducing more energy change at the receiver end, ...”.

✓ We used “sensitivity enhancement” and “amplification of the acoustic response” in the revised paper.

6. The authors apply inconsistent wording for a same thing. For example, they used “laser sensor” and “optical sensor” mixedly.

✓ Actually, they are not the same thing. The laser sensor presented in this work relies on the measurement of lasing phase change due to the ultrasound. “Optical sensors” include the laser sensor and also the existing sensors based on a passive optical resonator.

7. This reviewer recommends the authors to move the new insertion in Line 206 to a proper place in a supplementary figure because at the current location, it breaks the flow of more important story.

✓ According to this suggestion, the sentence at Line 206 was moved to the supplementary notes.

8. I wonder whether the section head “Conclusion and discussion” is reasonable. As far as this reviewer knows, “discussion-conclusion” or “summary-discussion” are more general structures when wrapping up a paper. Do you think suggesting a discussion after you already making a conclusion is really reasonable? In the current story elements, I think just going with “Discussion” looks enough.

- ✓ We have changed the heading to "Discussion", which is also required by the editors.

9. In terms of this kind of structurizing issue, this reviewer also suggests the authors to consider to move the story beginning from Line 246 to Line 261 to the end of the discussion section because it is related to a future study.

- ✓ According to the suggestion, we moved the discussion on technological limitations (Lines 246 to 261) to the end of the main text.